# SAX and Random Projection Algorithms for the Motif Discovery of Orbital Asteroid Resonance Using Big Data Platforms

**DOI:** 10.3390/s22145071

**Published:** 2022-07-06

**Authors:** Lala Septem Riza, Muhammad Naufal Fazanadi, Judhistira Aria Utama, Khyrina Airin Fariza Abu Samah, Taufiq Hidayat, Shah Nazir

**Affiliations:** 1Department of Computer Science Education, Universitas Pendidikan Indonesia, Bandung 40154, Indonesia; 2Department of Physics Education, Universitas Pendidikan Indonesia, Bandung 40154, Indonesia; 3Faculty of Computer and Mathematical Sciences, Universiti Teknologi MARA Cawangan Melaka Kampus Jasin, Melaka City 77300, Malaysia; 4Astronomy Research Division, Faculty of Mathematics and Natural Science, Institut Teknologi Bandung, Bandung 40132, Indonesia; 5Department of Computer Science, University of Swabi, Swabi 94640, Pakistan

**Keywords:** big data, SAX algorithm, random projection algorithm, time series, motif discovery, mean motion resonance

## Abstract

The phenomenon of big data has occurred in many fields of knowledge, one of which is astronomy. One example of a large dataset in astronomy is that of numerically integrated time series asteroid orbital elements from a time span of millions to billions of years. For example, the mean motion resonance (MMR) data of an asteroid are used to find out the duration that the asteroid was in a resonance state with a particular planet. For this reason, this research designs a computational model to obtain the mean motion resonance quickly and effectively by modifying and implementing the Symbolic Aggregate Approximation (SAX) algorithm and the motif discovery random projection algorithm on big data platforms (i.e., Apache Hadoop and Apache Spark). There are five following steps on the model: (i) saving data into the Hadoop Distributed File System (HDFS); (ii) importing files to the Resilient Distributed Datasets (RDD); (iii) preprocessing the data; (iv) calculating the motif discovery by executing the User-Defined Function (UDF) program; and (v) gathering the results from the UDF to the HDFS and the .csv file. The results indicated a very significant reduction in computational time between the use of the standalone method and the use of the big data platform. The proposed computational model obtained an average accuracy of 83%, compared with the SwiftVis software.

## 1. Introduction

Nowadays, almost everyone creates or uses new data every day, so the growth of data is unstoppable. For example, according to one study, Facebook uploads more than 250 million photos daily and handles interactions between 800 million active users and more than 900 million objects (pages, groups, etc.) [1]. Thus, this phenomenon has given rise to new terminology, namely big data. Big data is a term that represents the phenomenon of data, namely high-volume, high-speed, and diverse information assets that demand cost-effective information processing and innovative forms of information processing for improved insight and decision making [2]. This big data phenomenon does not only occur in computer applications but also in other fields. For example, researchers might attempt to solve a problem in biology by analyzing the repetition in a DNA sequence [3]. Another example is astronomy, where massive amounts of data are produced from ground- and space-based observations. So, analyzing the data using traditional computation methods is almost impossible. One of the forms of data that is particularly abundant in the field of astronomy is asteroid orbital time series data, which involves six elements in the Keplerian component, namely a (semimajor axis), e (eccentricity), i (inclination), ω (arguments of perihelion), Ω (longitude of the ascending node), and M (mean anomaly) [4]. Moreover, big data analysis is also used in agriculture [5]. This is because sensors/tools in agriculture produce massive datasets, such as weather and climate change, remote sensing, crop images, land conditions, etc.

In astronomy, there is a population of asteroids between the orbits of Mars and Jupiter. One contributing mechanism to delivering asteroids in the main belt into the near-Earth region could be high-velocity collisions between asteroids [6]. While in the near-Earth region, defined as a region that satisfies the inequality q (perihelion distance) < 1.3 au and Q (aphelion distance) > 0.98 au, asteroids can experience close encounters with certain planets (Mercury, Venus, Earth, and Mars) and also the moon. As a result of these close encounters, asteroids can fragment due to strong tidal forces, or if they survive, their orbits can change drastically in a short time. The drastic changes in orbits can change the future fate of asteroids, namely whether they still orbit the sun, collide with massive objects, or are ejected from the solar system. According to Gallardo [7], it is laborious to identify which one of the hundreds of mean motion resonances (MMRs) that theoretically exist near the semimajor axis of the orbit we are studying is the one affecting the asteroid’s motion. MMRs occur when two bodies have periods of revolution with a simple integer ratio, either stabilizing or destabilizing the orbit. Stabilization may happen when the two bodies move in such a synchronized fashion that they never closely approach. For small bodies, such as asteroids, destabilization is far more likely. Locations of MMRs are simply calculated with the use of Kepler’s third law, and the critical angle is used to refer to the actual resonance state.

Therefore, this research is focused on building a computational model by modifying the motif discovery algorithm on a big data platform and adapting its implementation so that it can provide motif search results and resonance results in a 1:1 state found in the asteroid orbital resonance time series data accurately and quickly. In our study, we use only the commensurability of the orbital period of an asteroid to the planets as an indicator for the existence of MMRs. We focus only on finding 1:1 MMR states among near-Earth asteroids with terrestrial planets and the moon. An asteroid can be in the 1:1 MMR with a given planet when |*a* − *a*_1_ | < (*μ*/3)^1/3^*a*_1_ is satisfied (*μ* is the planet–Sun mass ratio, whereas *a* and *a*_1_ are the asteroids and planet’s orbital semimajor axes, respectively). In this case, the orbital period of the asteroids around the sun are the same as the planets. Moreover, this study uses the two following algorithms for detecting motifs that occur in time series data without any motive input from the user: the SAX algorithm [8] used to change the time series representation, and the random projection algorithm [9] used to discover a motif. Using the SAX algorithm, a time series dataset can quickly change to string representation. Then, this is followed by the random projection algorithm to detect the motif data found in the data. Random projection can detect motifs without the need for the motif input data that users want to find quickly. To process the data, this research will use Apache Hadoop [10] and Apache Spark [11] as big data platforms and use the modified SAX algorithm and random projection algorithm to process big time series data. The Hadoop Distributed File System (HDFS) of Apache Hadoop will be used as data storage for storing large data and spread over several cluster nodes. Then, Apache Spark will be used as a computing medium for this research to process large data quickly by dividing the process into several nodes in the cluster. By using these platforms running in parallel and distributed computing modes, computational cost can be reduced while still maintaining reasonable accuracy.

In this study, the SAX algorithm and the random projection algorithm are implemented to solve motif discovery issues [12]. However, this research was conducted in a single processor. This means that the proposed method in this research is an improvement of the previous one. The other study involving Apache Spark on discovery patterns was performed by Jiang et al. [13], Riza et al. [14], and Pérez-Chacón et al. [15].

## 2. Research Methods

Figure 1 shows the computational model developed in this research. It involves two two big data platforms: Apache Hadoop YARN schedulers and Apache Spark. In Apache Hadoop, we used the Hadoop Distributed File System (HDFS) for storage and Apache Spark for the computation of the model. Therefore, it can be seen that this computational model was built to adopt and modify processes run in the standalone mode so that it could be run on a big data platform in parallel and distributed computing modes.

Basically, the computational model can be divided into the 4 following system environments: (i) in personal computers/local machines; (ii) virtual machines on the Google Cloud Platform [16]; (iii) the HDFS of Apache Hadoop; and (iv) computation using DataFrame in Apache Spark. Moreover, in these environments, there are four steps to be completed, as follows.

### 2.1. Data Collection

In the first step, we collect data in the form of the resonance data of asteroid orbital elements. The input data are uploaded first to the virtual machine on the Google Cloud Platform. Then, the files in the virtual machine are copied using the put command owned by Apache Hadoop into HDFS for processing the data in a cluster. After that, the file will be imported/used by Apache Spark. The data type used in the Apache Spark environment is a DataFrame containing a Resilient Distributed Dataset (RDD) [17] that has been partitioned into sections on each block file in HDFS.

### 2.2. Data Preprocessing

The first stage conducted in the Apache Spark environment is data preprocessing. Before completing the processing and calculations, we need to clean the data first. At this stage, preprocessing of the time series data input is carried out. This preprocessing is the stage of cleaning the raw data. After that, the next stage is the data normalization stage. The time series data are normalized using *Z*-score normalization so that all data are comparable for processing. The reason for using this normalization is because time series data tends to have data with a normal distribution [18]. The normalized time series data will then be converted into SAX and generate string data to be processed by the motif discovery random projection algorithm. The SAX method allows time series data to be converted into a string with the desired number of characters or alphabets in the string. The size of the alphabet is an arbitrary integer *a*, where *a > 2*. This algorithm consists of 2 steps: (i) the transformation of the initial time series into a time series with Piecewise Constant Model (PAA) representation [19]; and (ii) reducing the dimensions by converting PAA into a symbolic representation of the time series in the form of a string [20]. The process of the SAX algorithm is described in pseudocode in Figure 2 and an example of the results of the time series discretization is shown in Figure 3.

### 2.3. Computing with Motif Discovery in Big Data Platforms

After running the SAX algorithm and obtaining the string data output from the SAX algorithm, the next step is to find the previously unknown motif using the random projection algorithm [21]. This algorithm is taken from the problem of Planted Motif Search (PMS) in the field of computational biology, where PMS aims to find all the motifs that appear in each DNA sequence [22].

Random projection is one of the algorithms used for the problem of finding motifs in DNA sequences. In this algorithm, pieces of input data in the form of sub-sequences (*l-mers*) are projected according to a random position determined based on the value of *k* (*k-mers*) [23]. Random projection represents those mutations that can occur anywhere so random projections are performed randomly. The pseudocode of this algorithm can be seen in Figure 4 [9].

The next stage is the motif discovery stage using the SAX algorithm and the random projection algorithm designed and explained in the previous step. It should be noted that this computation is run as a Resilient Distributed Dataset (RDD) of Apache Spark in each data partition.

At this stage, in addition to the motive discovery function, there is a data normalization process and a data postprocessing stage which include several elements, such as combining the motifs found in a row, looking for values at the distance the AU motif is found, and filtering which locations have 1:1 mean motion resonance. The four functions use functions built in standalone Python; however, in Spark SQL, ordinary functions cannot be used for data processing for partitioned data frames. In this stage, there is a process of modifying the program code that has been made previously. The modification of the program code involves changing the three processes in the discovery motif into the User-Defined Function (UDF) in Apache Spark. This process is intended so that functions that have been created in standalone Python can be used in the Apache Spark and Spark SQL environments.

The UDF’s role is to run the commands it contains against all RDD data frame partitions that Apache Spark has split. For example, DataFrame x has 10 data divided into 4 partitions split by Apache Spark. Then, when the UDF, which contains data normalization, the SAX algorithm, random projection, and preprocessing, is executed, the three functions are executed simultaneously to each partition. To declare this UDF function requires a data type structure for the output of this function, namely an array in an array containing an array of integers which will contain the motif’s start location, the motif’s ending location, and a 1:1 resonance condition of the motif. The application of the UDF function and the modification of the algorithm are shown in Figure 5. It can be seen that, basically, the UDF program involves the following functions: *Z*-score normalization, SAX, random projection, and postprocessing.

The next step is to execute the previously created UDF function on the RDD data frame. Running UDF on a DataFrame cannot be used in the same way as entering data into a function as usual. The preprocessed DataFrame will apply the groupBy function first to combine all data rows with the same asteroid ID. Then, the data in column a will be collected in the same list. Next is data aggregation by applying and executing the created UDF function. The UDF function will be applied to each data partition and to each data list of column a that has been grouped together. After the UDF function is executed, the results will be stored in a new column and a new data frame containing the results obtained after performing the 4 main computational processes in the discovery motif, namely normalization, SAX, random projection, and the postprocessing of the data. After that, postprocessing will be carried out to process the results of the random projection into information that the user needs. This postprocessing includes several elements: combining the motifs found in a row, identifying the differences in the location of the motifs and looking for values at the distances at which the motifs are found, and filtering at any locations that experience 1:1 mean motion resonance. The output produced is in the form of information on the location of the motif, information of any time the motif is found, and at what distance the motif is found in the time series data.

### 2.4. Copy DataFrame to Local File (.csv)

Then, a new DataFrame that has a column resulting from the motif discovery process will be exported into a folder containing .csv. The resulting DataFrame that was previously split into several partitions will be combined first using the resulting file, and will be saved into the HDFS because the data will still be divided into several block files in different nodes. Once saved to the HDFS, the resulting folder is copied back into the virtual machine environment on the Google Cloud Platform. In this environment, the data are back to normal in the form of a regular .csv and the file is not broken into several parts in the local machine. The last step is to download the processed folder from the virtual machine to a personal computer.

## 3. Experimental Study

The calculation of the orbits of celestial bodies in astronomy, in the simplest form, involves two isolated bodies (the two-body problem), that is, one body of less mass orbits and another body of greater mass under the influence of their mutual gravitational attraction. Indeed, the two bodies orbit a common center of mass, where each object’s velocity and orbital distance from the common center of mass is determined by the mass of each object and its center-to-center distance. In the case of more than two objects (commonly known as the *N*-body problem), the same equation of motion can be extended to the number of simulated objects [4].

Identifying the MMRs affecting the asteroid’s motion is difficult due to the absence of a simple method that adequately weighs the strength of each resonance [7], or when several planets and a large number of asteroids are to be considered [8]. Based on the geometrical meaning of the resonance variable, Forgács-Dajka et al. [8] introduce an efficient method by which MMRs can be easily found without any a priori knowledge of them. Our study simplified the problem into 1:1 MMR for two-body (planet and asteroid) consideration using a big data analysis platform.

### 3.1. Data Collection

The data used in this study are the time series data of asteroid orbital element resonance. The data are on the evolution of the orbits of celestial bodies in the form of expected asteroids over the next several million years. An example of the time series dataset of the asteroid orbital element resonance can be seen in Figure 6. The data used were obtained from initial sources in the JPL NASA Small Body Database Search Engine (http://ssd.jpl.nasa.gov/sbdb_query.cgi; accessed on 3 March 2016) by filtering only for NEOs (near-Earth objects), which are asteroids of four classes (Amor, Apollo, Aten, and Atira), both numbered and unnumbered with very well-known orbits. As of 3 March 2016, 3372 NEAs (near-Earth asteroids) were obtained according to the epoch MJD57400. The data obtained comprised four asteroid orbital resonance data files, which can be seen in Table 1.

Table 1 shows that the total files used were four .txt files. Each file had a different maximum duration of orbit computation, a different file size, and a different number of lines. The total number of 3372 real asteroids was used and distributed across four files where one file consisted of a maximum of 1000 asteroids. The total number of lines from the four files was 12,693,476 lines with a total file size of 977 MB. In the file name, there was information about the duration of computation. For example, the first .txt file had the longest duration of orbit computation: up to 20 million years in the future from the current epoch. The number of asteroids in each file will also decrease with time. Different processes such as asteroid collision with the sun, planets, or ejection from the solar system can cause decay.

Table 2 shows the data used in this research experiment. Due to space limitations, the author only shows the first 19 rows of 10 asteroids out of 1000 asteroids in the first batch of data. In this data, each asteroid is sampled every 1000 years and stored in days. The number of columns in this dataset is 8, namely *AsteroidID*, *t*, *a*, *e*, *i*, *OMEGA*, *omega*, and *M* (representing the asteroid ID, time, semimajor axis, eccentricity, inclination, argument of perihelion, longitude of the ascending node, and mean anomaly).

In this study, only the first three columns were used, namely *AsteroidID*, *t*, and *a*, because this study focuses on finding the location and time of the occurrence of the found MMRs. In the *AsteroidID* column, the data with a negative value represent the planet being considered for MMRs. These planetary data are used as a reference of 1:1 mean motion resonance with asteroids. The data are sorted in ascending order by column *t*, containing the time sampled from the computation process. The unit of time in column *t* is in days. At each multiple, this column is added by 365,250 days which is equal to 1000 years. Then, column *a* has a unit value of AU, which means the average distance between the asteroid and the sun. Column *a* is the time series data that will be processed for the discovery motif because it can determine any 1:1 mean motion resonance occurrence in each asteroid.

### 3.2. Experimental Scenario

In conducting the experiment, we first designed the scenario for the experiment. The parameters used in all experiments were the same. The following are the details of the design of the parameters for both experiments:*sax_cuts* = 20. This was used as a parameter to enter the number of letters used in the SAX algorithm. The parameter *sax_cuts* was filled with the number 20 because this is the largest number of letters that can be specified in SAX.*l_find* = 10. This was used to determine the minimum length of the motif sought in the random projection algorithm. This parameter contained a value of 10 because we identified a resonance in the asteroid orbital data if the asteroid had a minimum length of 10,000 years or 10 time units in the time series.*Mismatch* = 1. This was used to determine the minimum number of mismatches that occurred in the motifs in the random projection algorithm.*n_try* = 1. This was used to determine the number of trials or iterations that were carried out to run the random projection algorithm.*threshold* = 2. This was used to determine the minimum number of the same motifs found so that the collection of motifs was called resonance by the user. The author gives a value of 2 because this was the minimum number of buckets to be called a motif.*threshold_planet* = 0.05. This was used to determine the maximum and minimum limits of the 1:1 resonance of asteroids with each planet from the distance value on each planet.

We performed two experiments. First, we experimented with using multiple worker nodes, with each node having four CPU cores; the second experiment involved using multiple CPU cores on two worker nodes on the Google Cloud Platform. In the first scenario, the author experimented on a cluster that used several worker nodes, with each node having four CPU cores, as can be seen in Table 3.

Table 3 shows that the experiment used different worker nodes and cores on the same nodes on the Google Cloud Platform. In the experimental scenario using 1 master and 0 worker nodes, the computation was run standalone with the master only, so the master node worked as a worker node. In the other experimental scenarios, the master did not perform the computations.

In the second scenario, the author experimented on a cluster that used several CPU cores on two worker nodes, as can be seen in Table 4.

Table 4 shows that the experiment used different numbers of cores and the same worker node on the Google Cloud Platform. Unlike the case with the first experimental scenario, the meaning of the number of cores here is that the cores used in two worker nodes were added up. For example, if a worker node had four CPU cores, the number of cores was eight cores.

It should be noted that Table 5 shows descriptions of the hardware specifications that were used in the experiments.

## 4. Results and Analysis

After conducting two experimental scenarios, the results are presented in the following section.

### 4.1. Experimental Results with Four Cores and Various Worker Nodes

The results of the first experiment can be seen in Table 6. The ‘Cost Time’ column shows the time to complete each computation in minutes for all files. The ‘Speedup’ column is a performance measure. This column measures the ratio of execution time and execution time on a cluster-per-node basis. Then, the ‘Efficiency’ column is a measure of the use of the computing resources. It measures the ratio between the performance and the resources used to achieve that performance.

The experimental results of the four cores with several worker nodes showed a significant comparison of computing time, as shown in Figure 7. From Figure 7 and Table 6, it can be seen that the more workers used in the cluster the faster the computing process was. This shows that the number of nodes used significantly affected the computational speed. The longest time taken for processing data was on the 977 MB file, which was 974.23 min or equivalent to 16.23 h using 1 worker node, while the fastest time was 78.28 min or equivalent to 1.3 h using 15 workers nodes, which had a speedup value of 12.4 times faster than using 1 worker node.

The results of this experiment are interesting: as the worker nodes increased, the increase in computational speed was not proportional to the number of worker nodes added. This is shown in the efficiency column: the more worker nodes used the lower the efficiency was. For example, in the experiment with 10 worker nodes, the speed increase was not two times faster than an experiment that used 5 worker nodes. Similarly, when using 10 worker nodes to 15 worker nodes, the speed increase was not the same as using 5 workers to 10 workers. The increase in speed from 5 workers to 10 workers was 85.22 min, while the increase in speed from 10 workers to 15 workers was 26.79 min.

### 4.2. Experimental Results with Various Cores and Two Worker Nodes

The results of the first experiment can be seen in Table 7. The ‘Cost Time’ column shows the amount of time to complete each computation in minutes for all files. The ‘Speedup’ column is the performance measure. This column measures the ratio of execution time and execution time on a cluster-per-four-cores basis. Then, the ‘Efficiency’ column is a measure of the use of computing resources. It measures the ratio between performance and the resources used to achieve that performance.

The experimental results with various cores and two worker nodes showed a significant comparison of computing time, as shown in Figure 8. From Figure 8 and Table 7, it can be seen that the more total cores used on two worker nodes, the faster the computational process. This shows that the number of cores used significantly affected the computing speed. The longest time taken for processing the data was for the 977 MB file, which took 866.97 min or equivalent to 14.44 h using 4 cores, while the fastest time was 87.73 min, or equivalent to 1.46 h, using 48 cores on two worker nodes. This had a speedup value of 9.8 times faster than using four cores.

Similar to the experiment using various worker nodes and four cores, the results of this experiment were interesting: as the number of cores increased, the increases in computing speed were not proportional to the number of cores added. This is shown in the efficiency column, which shows that the more cores used the lower the efficiency was. For example, when the experiment used 32 cores, the speed increase was not two times faster than the experiment using 16 cores. Likewise, when using 32 cores to 48 cores, the speed increase was not the same as using 16 cores to 32 cores. The speed increase from using 16 cores to 32 cores was 99.69 min, while the speed increase from 32 cores to 48 cores was 42.58 min.

### 4.3. Speed Comparison with SwiftVis Software

In terms of computation time, this study compared speed using SwiftVis software (available at https://www.cs.trinity.edu/~mlewis/SwiftVis/; accessed on 18 February 2019), which is a tool for finding MMRs commonly used in planetary science research. To obtain the MMR states from the SwiftVis application, users are required to input the time series data of orbital computations, choose a selection function, add a filter function, and choose a general plot function to obtain any asteroids in an MMR state with a particular planet, as well as the duration of the MMR state with the planet. SwiftVis output can be saved as a new .txt file. The speed comparison for all data with the first scenario against the SwiftVis software is shown in Table 8 and the speed comparison for the second scenario is shown in Table 9.

According to the speed comparison shown in Table 9, the SwiftVis software had a slower time than the research results. The time required to process all data with SwiftVis was about 9052.08 min, or equivalent to 6.2 days, while the longest time required by this research was only 974.23 min. This comparison’s results showed that this research was much faster, even with the use of one worker node to process data, compared to the SwiftVis software.

The speed calculation in SwiftVis software starts from entering data and involves manually recording asteroids that experience 1:1 resonance. Speed results with SwiftVis are approximate data. This is because if they are processed whole this would take a lot of time and energy. Speed calculations were generated from calculations against one SwiftVis output file. The speed obtained was 10 min for data that had 1411 rows of data. If 1411 rows can be processed in 10 min, then 1 row can be processed in 0.00709 min. Then, the speed for processing one line of data was multiplied by the total number of output data lines, which produced a total speed of 9007.02 min. After being multiplied, this was then added to the speed of the SwiftVis software, which generated output data at a speed of 45.06 min. The two-process speed results were then added up to 9052.08 min.

### 4.4. Accuracy Comparison with SwiftVis Software

After the experiments were carried out, the authors compared the output of the program that was built with the SwiftVis software to ensure that the output of the program matched the results issued by SwiftVis. Table 9 describes the comparison of the results of the experiments carried out and the output of the SwiftVis with the same *threshold_planet* parameter of 0.05 and the minimum resonance length of the *l_find* parameter, which was 10 units of time. In the results of the SwiftVis software, which could not determine the minimum limit of the resonance length found, the results of a resonance location that was in the 1:1 threshold range, even though only one unit of time was entered into the *planet_threshold* parameter, were produced. In this case, all results whose time range was less than the parameter *l_find*, which was 10, were considered to have no resonance in that time range.

Due to a large amount of data, the authors took a sample of 10 asteroids from 3372 asteroids. This sample consisted of the first 10 asteroids and 20 asteroids that were taken randomly from the orbit integrator package’s output file to determine the built program’s accuracy. In the SwiftVis output column, if there were asteroids in resonance with certain planets, then the column was divided into two parts at each point. In the first part was a description of which planet the asteroid resonated 1:1. The second part described the year the asteroid experienced 1:1 resonance in 1000-year units. A comparison of results is shown in Table 10. Based on the results of the output comparison table, the built program obtained an average accuracy of 83% from 30 asteroid asteroids with the minimum required resonance length according to the input *l_find* in the scenario, which was 10. Accuracy was obtained based on the percentage of the same number of time units between the experimental results and the SwiftVis results. To calculate the accuracy percentage, all experimental results of the slices’ resonance time ranges were combined first and then compared for matches with the results issued by SwiftVis. For example, the asteroid with *ID* 6 had two resonances between 71–82 and 76–86. Because the two time ranges had a wedge from index 76 to 82, this was combined into a new time range of 71–86. Then, the time range was compared with the SwiftVis output that had the same timeframe. Then, it was found that the accuracy obtained for the asteroid with *ID* 6 from the experimental results was 100%.

There was tolerance in the calculation of accuracy. If the program produced an output that did not exist in the SwiftVis output, the results were ignored and did not affect the accuracy percentage. For example, the asteroid with *ID* 3 showed a 1:1 resonance with the moon, but in the SwiftVis output the asteroid did not have a 1:1 resonance with the moon. Tolerance was also applied if the time range generated by the program exceeded the time range generated by SwiftVis. It was ignored and did not affect the accuracy. In the output of the experimental results, the asteroid with *ID* 3 had a 1:1 resonance with the Earth in the time range 69–125, while in SwiftVis it had a time range of 94–122. So, the experimental results were considered to be suitable because they included the results from the SwiftVis output, even though the time span exceeded the range result time from SwiftVis. An error in the program occurred in the result of the sample asteroid with *ID 7*, which should have had a 1:1 resonance with Mars in the time range of 211 to 234. This was due to the different methods used in the two programs. SwiftVis only uses selection and filtering, but the program we built uses motif discovery, so it is possible to obtain different resonance locations. Then, there is the calculation of the location of the motifs that uses the average *a*-value of the motifs found; the *a*-values in the motifs found exceeded the threshold and were thus not included in the 1:1 resonance.

## 5. Conclusions

The main contributions of this research are: (i) providing a computational model for time series motif discovery with a big data platform in the case of mean motion resonance; (ii) from the experimental results, it can be concluded that the more worker nodes or the more cores used will significantly speed up the computing process, but using more worker nodes or cores will not guarantee an increase in the efficiency of the use of worker nodes and cores in computing; (iii) the determination of the accuracy of the program that has been built. Our method obtained an average accuracy of 83% of a sample of 30 asteroids from 3372 asteroids, compared with the SwiftVis software. In the future, we have plans to improve this system’s accuracy and processing speed, and it is hoped that further research can produce specific information for those asteroids in 1:1 MMR with the Earth. These objects could become a destination for space mineral mining, and are thus of economic value.

## Figures and Tables

**Figure 1 sensors-22-05071-f001:**
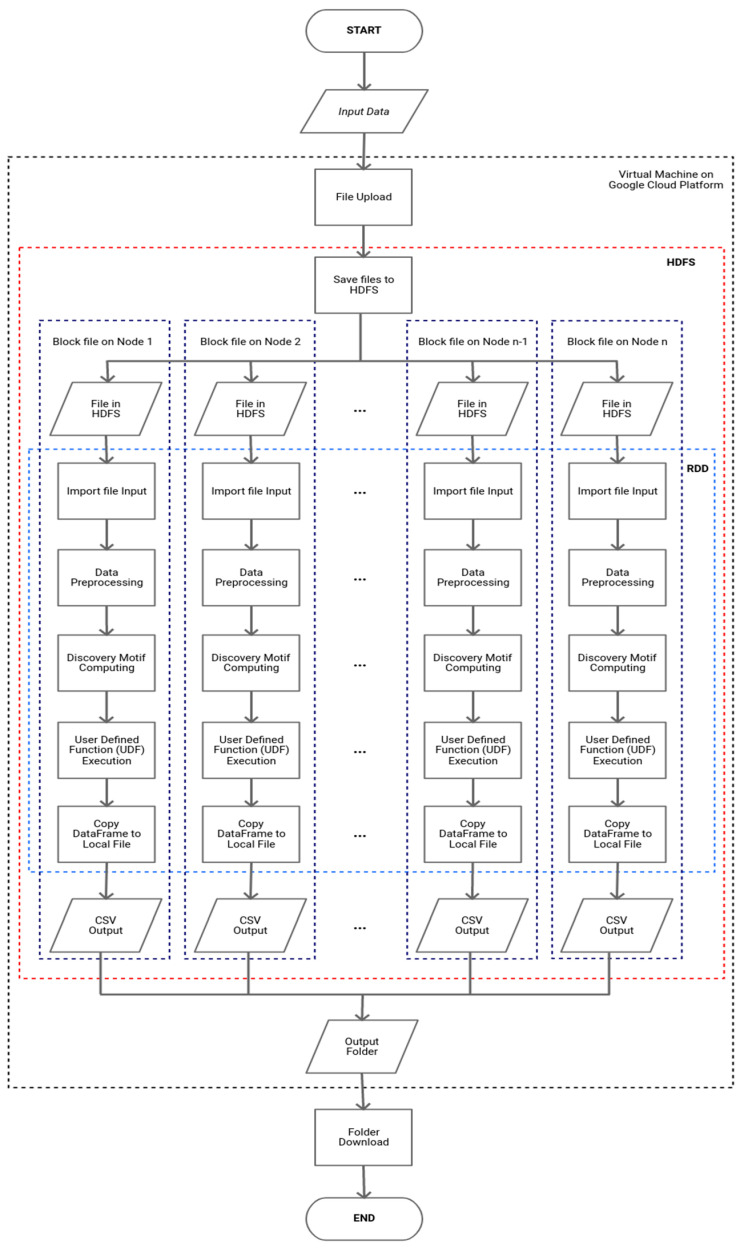
Computational model in the cluster engine using a big data platform.

**Figure 2 sensors-22-05071-f002:**
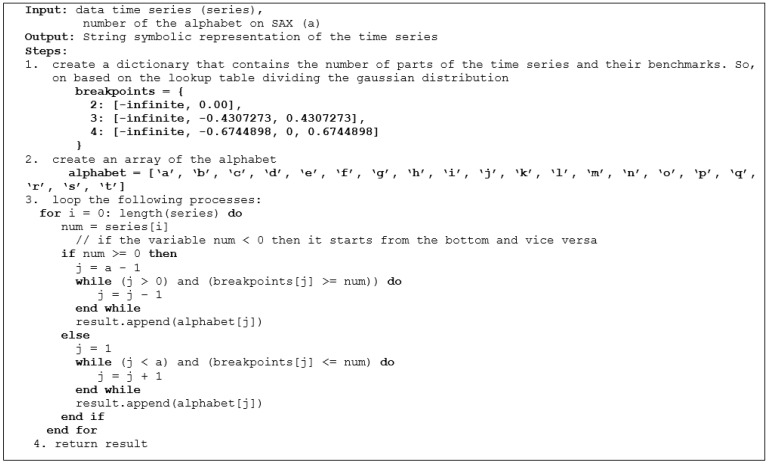
Pseudocode of the SAX algorithm in the standalone mode.

**Figure 3 sensors-22-05071-f003:**
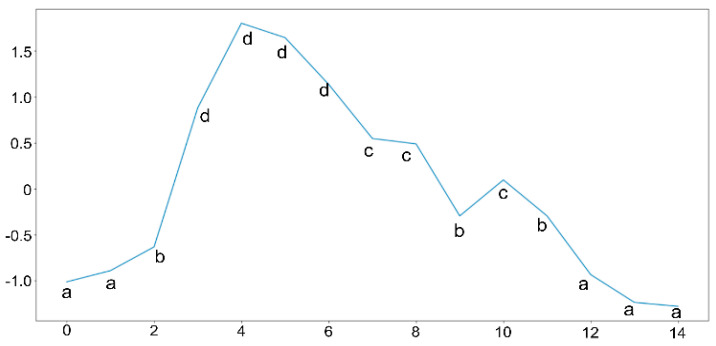
Time series that have been discretized to be labels of SAX in the following string “aabddddccbcbaaa”.

**Figure 4 sensors-22-05071-f004:**
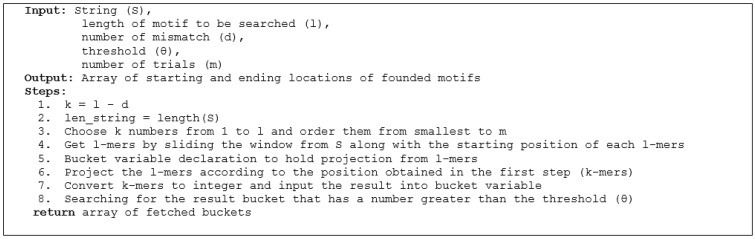
Pseudocode of the random projection algorithm.

**Figure 5 sensors-22-05071-f005:**
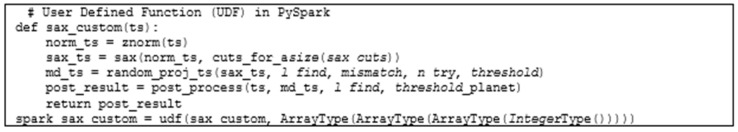
The command of SAX and random projection in the UDF of Spark SQL.

**Figure 6 sensors-22-05071-f006:**
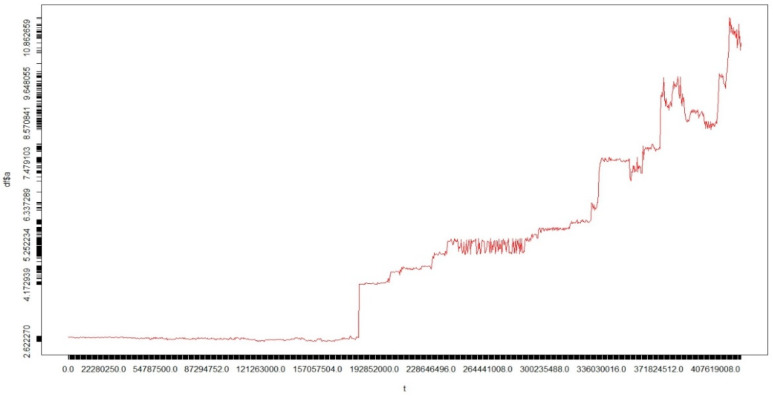
An example of a time series dataset of asteroid orbital element resonance.

**Figure 7 sensors-22-05071-f007:**
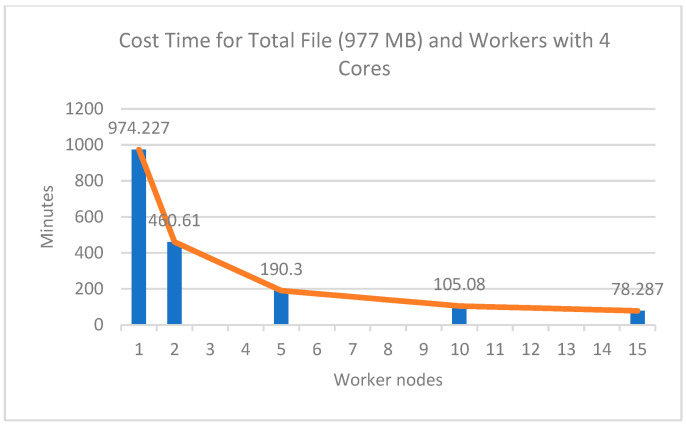
Cost time profile between workers against time.

**Figure 8 sensors-22-05071-f008:**
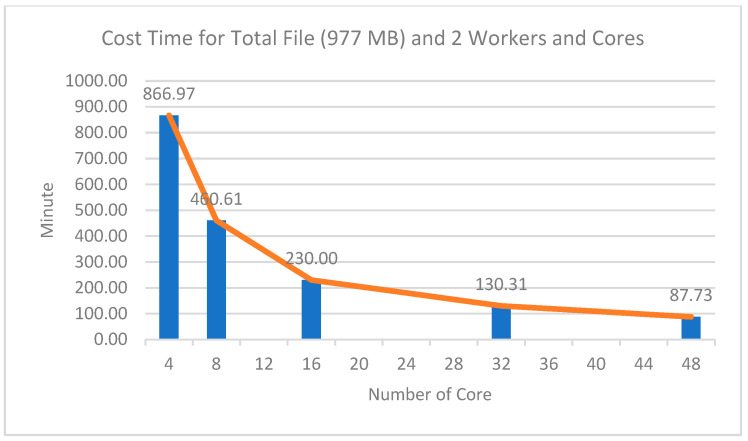
Cost time profile between total cores and time.

**Table 1 sensors-22-05071-t001:** List of resonance data of asteroid orbital elements.

File Name	File Size (MB)	Number of Lines	Number of Asteroids
Batch1_1000Asteroid_20JutaTahun.txt	359,355	4,660,333	1000
Batch2_1000Asteroid_5JutaTahun.txt	310,874	4,033,633	1000
Batch3_1000Asteroid_8JutaTahun.txt	176,910	2,303,610	1000
Batch4_372Asteroid_10,5JutaTahun.txt	130,156	1,695,900	372
Total	977,295	12,693,476	3372

**Table 2 sensors-22-05071-t002:** Examples of data used in experiments.

*AsteroidID*	*t*	*a*	*e*	*i*	*OMEGA*	*Omega*	*M*
−2	0.0	0.3870980	0.2056274	0.1222434	0.843181	0.5090891	0.26909
−3	0.0	0.7233265	0.0067518	0.0592433	1.3375275	0.9612226	1.23472
−4	0.0	1.0007949	0.0174236	0.0000602	2.530922	5.5680704	0.12554
−5	0.0	0.9405935	0.0478169	0.0027293	5.115822	0.38571003	2.68625
−6	0.0	1.523777	0.0933678	0.0322604	0.86408144	5.002638	3.62185
−7	0.0	5.2025285	0.0488403	0.0227536	1.7542297	4.779151	2.55758
−8	0.0	9.556443	0.0533448	0.0434584	1.9813833	5.9391503	2.65307
−9	0.0	19.14179	0.0497408	0.0134874	1.2896736	1.7050518	3.67267
−10	0.0	29.975988	0.0075758	0.0308184	2.2987971	5.122956	4.79311
1	0.0	1.061678	0.051489	0.022113	4.421005	5.529237	2.02073
2	0.0	1.942493	0.555555	0.079524	4.411401	4.030324	3.01016
3	0.0	0.830664	0.388456	0.088272	2.900785	5.581375	4.84435
4	0.0	0.988711	0.013889	0.075412	4.614986	1.527315	3.08849
5	0.0	2.085440	0.606626	0.032093	4.734172	1.805200	0.44300
6	0.0	0.996808	0.012064	0.144113	0.434161	3.888507	3.87216
7	0.0	1.210198	0.177278	0.134907	2.571753	0.872216	5.25321
8	0.0	1.007883	0.139036	0.038552	3.671272	5.048894	5.85180
9	0.0	1.306163	0.246871	0.075872	5.275945	5.695753	0.27101
10	0.0	0.992638	0.083851	0.013485	1.455619	2.622877	4.49503
…	…	…	…	…	…	…	…
957	7.304 × 10^9^	1.5908555	0.0889270	0.3030403	3.3423336	1.8881923	1.93854
961	7.304 × 10^9^	1.9744761	0.4365535	0.5805287	3.2029006	5.186608	2.33091

**Table 3 sensors-22-05071-t003:** Experimental scenario with four cores and various worker nodes.

No	Master	Worker Nodes	Core per Node
1	1	0	4
2	1	2	4
3	1	5	4
4	1	10	4
5	1	15	4

**Table 4 sensors-22-05071-t004:** Experimental scenario with various cores and two worker nodes.

No	Master	Worker Nodes	Number of Core
1	1	2	4
2	1	2	8
3	1	2	16
4	1	2	32
5	1	2	48

**Table 5 sensors-22-05071-t005:** Description of hardware specification in the experiments.

Mode Description	Hardware Specification
Standalone	Processor Intel^®^ Core™ i7-8550U 8 Cores CPU Memory 8 GB RAM HDD 1 Tera Bytes
Cluster/Parallel and Distributed Computing	- Cluster with 1 namenode and 2 worker nodes, 5 worker nodes, 10 worker nodes, and 15 worker nodes in the experiment, as illustrated in Table 3: (a) Namenode specification: Processor Intel Broadwell 4 Cores, Memory 15 GB RAM, HDD 32 GB. (b) Worker node specification: Processor Intel Broadwell 4 Cores, Memory 15 GB RAM, HDD 32 GB. - Cluster with 1 namenode and 2 worker nodes with various cores in the experiment, as illustrated in Table 4: (a) Namenode specification: Processor Intel Broadwell 4 Cores, Memory 15 GB RAM, HDD 32 GB. (b) Worker node specification: Processor Intel Broadwell 2 cores, 4 cores, 8 cores, 16 cores, dan 24 cores, Memory 7.5 GB RAM, 15 GB RAM, 30 GB RAM, 60 GB RAM, dan 90 GB RAM, HDD 32 GB.

**Table 6 sensors-22-05071-t006:** Experimental results of four cores and various worker nodes (all files).

No	Master	Worker Nodes	Number of Core	Cost Time	Speedup	Efficiency per Node
1	1	0	4	974.23	1	1
2	1	2	4	460.61	2.11508	1.05754
3	1	5	4	190.30	5.119427	1.023885
4	1	10	4	105.08	9.271289	0.927129
5	1	15	4	78.29	12.4443	0.82962

**Table 7 sensors-22-05071-t007:** Experimental results of four cores and various worker nodes (all files).

No	Master	Worker Nodes	Number of Core	Cost Time	Speedup	Efficiency per 4 Cores
1	1	0	4	974.23	1	1
2	1	2	4	460.61	2.11508	1.05754
3	1	5	4	190.30	5.119427	1.023885
4	1	10	4	105.08	9.271289	0.927129
5	1	15	4	78.29	12.4443	0.82962

**Table 8 sensors-22-05071-t008:** Speed comparison of first scenario with SwiftVis.

No	Result of	Worker Nodes	Cost Time (Minute)
1	Research	1	974.23
2		2	460.61
3		5	190.30
4		10	105.08
5		15	78.29
6	SwiftVis	-	9052.08

**Table 9 sensors-22-05071-t009:** Speed comparison of first scenario with SwiftVis.

No	Result of	Worker Nodes	Cost Time (Minute)
1	Research	4	866.97
2		8	460.61
3		16	230.00
4		32	130.31
5		48	87.73
6	SwiftVis	-	9052.08

**Table 10 sensors-22-05071-t010:** Accuracy comparison of second scenarios and SwiftVis.

No	*AsteroidID*	System Experiment Results	SwiftVis Application Output	Accuracy
1	1	no 1:1 resonance	no 1:1 resonance	100%
2	2	no 1:1 resonance	no 1:1 resonance	100%
3	3	1:1 moon. 63–73 1:1 earth. 69–125 1:1 Mars. 212–226 1:1 Mars. 219–228 1:1 Mars. 231–240 1:1 Mars. 220–229 1:1 Mars. 232–241 1:1 Mars. 221–230 1:1 Mars. 233–242	1:1 Earth. 94–122 1:1 Mars. 204–238	88%
4	4	no 1:1 resonance	no 1:1 resonance	100%
5	5	no 1:1 resonance	no 1:1 resonance	100%
6	6	1:1 Mars. 71–82 1:1 Mars. 76–86	1:1 Mars. 71–86	100%
7	7	no 1:1 resonance	1:1 Mars. 211–234	0%
8	8	no 1:1 resonance	no 1:1 resonance	100%
9	9	1:1 Mars. 35–45 1:1 Mars. 38–58	1:1 Mars. 34–57	95%
10	10	no 1:1 resonance	no 1:1 resonance	100%
11	57	1:1 Mars. 59–99 1:1 Mars. 95–106 1:1 Mars. 99–152	1:1 Mars. 80–145	100%
12	74	1:1 Mars. 24–127	1:1 Mars. 64–115	100%
13	153	1:1 Mars. 59–70 1:1 Mars. 65–138 1:1 Mars. 132–143	1:1 Mars. 61–149	93%
14	251	no 1:1 resonance	no 1:1 resonance	100%
15	299	1:1 Earth. 39–100 1:1 Mars. 180–191	1:1 Earth. 66–93	100%
16	344	no 1:1 resonance	1:1 Mars. 832–893	0%
17	380	1:1 Mars. 557–568	1:1 Mars. 527–590	17.46%
18	403	1:1 Mars. 539–599	1:1 Mars. 560–598	100%
19	446	1:1 Mars. 78–266	1:1 Mars. 145–181	100%
20	462	1:1 Earth. 0–163 1:1 Mars. 1332–1440 1:1 Mars. 1434–1445 1:1 Mars. 1440–1588 1:1 Mars. 1582–1593 1:1 Mars. 1588–1755	1:1 Earth. 24–40 1:1 Earth. 70–161 1:1 Mars. 1348–1743	100%
21	558	no 1:1 resonance	no 1:1 resonance	100%
22	598	no 1:1 resonance	no 1:1 resonance	100%
23	646	no 1:1 resonance	no 1:1 resonance	100%
24	700	no 1:1 resonance	no 1:1 resonance	100%
25	727	1:1 Venus. 147–773 1:1 moon. 1099–1110 1:1 Earth. 1104–1369 1:1 Earth. 1364–1375 1:1 Mars. 2007–2057 1:1 Mars. 2052–2063 1:1 Mars. 2057–2254	1:1 Venus. 341–563 1:1 Earth. 1304–1401 1:1 Mars. 2029–2251	95.2%
26	799	no 1:1 resonance	no 1:1 resonance	100%
27	822	1:1 Mars. 15–26	no 1:1 resonance	0%
28	876	1:1 Mars. 41–52 1:1 Mars. 45–229	1:1 Mars. 45–228	100%
29	903	no 1:1 resonance	no 1:1 resonance	100%
30	975	1:1 Mars. 296–319 1:1 Mars. 310–320 1:1 Mars. 312–322	1:1 Mars. 318–330 1:1 Mars. 332–353	12.12%
Average accuracy:				83%

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
