# Peer review of "SAX and Random Projection Algorithms for the Motif Discovery of Orbital Asteroid Resonance Using Big Data Platforms"

_sensors, 2022, doi:10.3390/s22145071_

Round 1
Reviewer 1 Report
The proposed algorithms show cases that Spark and Hadoop computing models can be easily adopted in many scientific fields including astronomy. Presented experiments lack appropriate description of system specification and configuration which makes comparing the efficiency of the algorithm impossible. Authors have not provided good reference points to compare Spark with other available solutions. Minor revision of the article is suggested.
It is not clear if computations were performed on one VM or multiple VMs. Figure 2. suggests that authors used only one machine. If more than one node was used: Are worker nodes homogenous or heterogenous? How is the Spark configured? Does it use Spark Standalone FIFO scheduler or one of the Hadoop YARN schedulers?
What is the configuration of storage on the virtual machines?
- Are the disks mounted from the server hosting the VMs? Is there a redundancy technique used that may influence computation speed (i.e., reading HDFS data from LVM logical volumes is notoriously slow)?
- If disks are mounted from external storage, then what is the point of using HDFS?
It is not clear if SwiftVis should be used as benchmark software in the field of Astronomy, are there any other similar solutions?
Authors use three different acronyms of User Defined Function: 1) UDP (line 23 and 25), 2) EDF (Figure 1.), 3) UDF (line 169, 172 etc.)
Line 201: DataFrame and RDD are different APIs.
Line 201: this is not how HDFS works. Did you mean: data is copied from HDFS to local storage?
Line 332: this is Amdahl’s law in action.
Line 372: What is the point of comparing SwiftVis (a GUI application that can run local tasks) with Spark (distributed processing model)? Can SwiftVis be run in parallel? Is it possible to run Spark tasks that will launch SwiftVis instances in batch mode (https://www.cs.trinity.edu/~mlewis/SwiftVis/BatchProcessing.html)? This way, authors could compare the speed of the proposed solution with competing software.
Authors use three different acronyms of User Defined Function: 1) UDP (line 23 and 25), 2) EDF (Figure 1.), 3) UDF (line 169, 172 etc.
Line 201: DataFrame and RDD are different APIs.
Line 201: this is not how HDFS works. Did you mean: data is copied from HDFS to local storage?
Line 332: this is Amdahl’s law in action.
Line 372: What is the point of comparing SwiftVis (a GUI application that can run local tasks) with Spark (distributed processing model)? Can SwiftVis be run in parallel? Is it possible to run Spark tasks that will launch SwiftVis instances in batch mode (https://www.cs.trinity.edu/~mlewis/SwiftVis/BatchProcessing.html)? This way, authors could compare the speed of the proposed solution with competing software.
Authors use three different acronyms of User Defined Function: 1) UDP (line 23 and 25), 2) EDF (Figure 1.), 3) UDF (line 169, 172 etc.)
Line 201: DataFrame and RDD are different APIs.
Line 201: this is not how HDFS works. Did you mean: data is copied from HDFS to local storage?
Line 332: this is Amdahl’s law in action.
Line 372: What is the point of comparing SwiftVis (a GUI application that can run local tasks) with Spark (distributed processing model)? Can SwiftVis be run in parallel? Is it possible to run Spark tasks that will launch SwiftVis instances in batch mode (https://www.cs.trinity.edu/~mlewis/SwiftVis/BatchProcessing.html)? This way, authors could compare the speed of the proposed solution with competing software.
The red underline can be removed from text in some figures. Source code should be provided as text, not as a picture?
Reviewer 2 Report
- The rationale for using Apache Hadoop and spark should be clearly provided.
- The contribution and motivation of the paper are not clear in the introduction. It should be written in the itemized form.
- Actually, the start of the introduction is fine with simple sentences. However, it has become very technical in the later paragraphs of the introduction. This is then difficult to understand.
- The rationale of the computation model given in Figure 1 should be explained clearly.
- Pseudocode given in Figure 2 should be simplified or clearly explained.
- Explain the steps of the computation model in a simple way and then go into details later. At this stage, it is difficult to understand.
- Table 2 provides the example of data. It is really making the paper difficult to read. Further, all the terms are not explained here.
- Figure 6 also gives the example of data. Is the figure drawn by yourself? If it is taken from other sources, please refer to it or draw it yourself.
- Tables 7 to 9 should be explained in a Figure format. Also, the explanation of the results should be improved.
- Part of the conclusion can be moved to the introduction and the whole conclusion can be re-written.
Round 2
Reviewer 1 Report
no
Reviewer 2 Report
All my comments are addressed. I would suggest accepting the paper.